# Nursing Diagnoses, Planned Outcomes and Associated Interventions with Highly Complex Chronic Patients in Primary Care Settings: An Observational Study

**DOI:** 10.3390/healthcare10122512

**Published:** 2022-12-12

**Authors:** Pedro-Ruymán Brito-Brito, Martín Rodríguez-Álvaro, Domingo-Ángel Fernández-Gutiérrez, Carlos-Enrique Martínez-Alberto, Antonio Cabeza-Mora, Alfonso-Miguel García-Hernández

**Affiliations:** 1Nursing Department, Faculty of Healthcare Sciences, Universidad de La Laguna, 38200 Santa Cruz de Tenerife, Spain; 2Canary Islands Research Group on Nursing Taxonomies (CARECAN), 38200 Santa Cruz de Tenerife, Spain; 3Primary Care Management Board of Tenerife, The Canary Islands Health Service, 38003 Santa Cruz de Tenerife, Spain; 4Health Services Management Board of La Palma, The Canary Islands Health Service, 38713 Breña Alta, Spain; 5Nuestra Sra, de la Candelaria Nursing College, 38010 Santa Cruz de Tenerife, Spain; 6Primary Care Management Board of Gran Canaria, The Canary Islands Health Service, 35006 Las Palmas de Gran Canaria, Spain

**Keywords:** needs assessment, standardised nursing terminology, chronic disease, long-term care, primary healthcare

## Abstract

The information logged by nurses on electronic health records (EHRs) using standardised nursing languages can help us identify the characteristics of highly complex chronic patients (HCCP) by focusing on care in terms of patients’ health needs. The aim of this study was to describe the profile of HCCPs using EHRs from primary care (PC) facilities, presenting patients’ characteristics, functional status based on health patterns, NANDA-I nursing diagnoses, health goals based on Nursing Outcomes Classification (NOC), and care interventions using Nursing Interventions Classification (NIC). With an observational, descriptive, cross-sectional, epidemiological study design, this study was carried out with a sample of 51,374 individuals. The variables were grouped into sociodemographic variables, clinical variables, resources, functional status (health patterns), nursing diagnoses, outcomes, and interventions. A total of 57.4% of the participants were women, with a mean age of 73.3 (12.2), and 51% were frail or dependent. Prevalent conditions included high blood pressure (87.2%), hyperlipidaemia (80%), osteoarthritis (67.8%), and diabetes (56.1%). The participants were frequent users of healthcare services, with 12.1% admitted to hospital in the past year. Some 49.2% had one to four health patterns assessed, with more information on biological and functional aspects than on psychosocial aspects. The mean number of nursing diagnoses was 7.3 (5.2), NOC outcomes 5.1 (4.1), and NIC interventions 8.1 (6.9). Moderately and highly significant differences were observed between dysfunction in physical activity/exercise health pattern and age group, and between dysfunction in other health patterns and classification as a frail or dependent elderly person. Regarding the presence of certain nursing diagnoses, significant differences were observed by age group, classification of elderly person status, and presence of diseases. A total of 20 NIC interventions showed moderately or relatively strong associations for older age groups, higher levels of dependency, and chronic health conditions.

## 1. Introduction

Chronic health conditions are common in our society and require adequate care planning. These conditions incur significant costs for healthcare systems and have a serious impact on the health and quality of life of patients and their caregivers. They require ongoing care and measures to enhance autonomy and provide health education in order to empower families.

Numerous changes must be made to the way in which healthcare systems are organised to enable the implementation of people-centred chronic care models that deliver satisfactory health outcomes. One of the most well-known models is the Chronic Care Model [1], which is used around the world. The model was developed in 1998 to identify populations at risk of poor outcomes.

The Chronic Care Model was modified by the World Health Organisation (WHO) in the document *Innovative Care for Chronic Conditions* [2], and by other authors in *The Expanded Chronic Care Model* [3]. The modified version covers health determinants and a series of coordinated interventions between different healthcare types, levels, and settings. The included measures go beyond clinical interventions to encompass health promotion, disease prevention, early screening and detection, case management, rehabilitation, and palliative care. 

Other initiatives with a similar aim include care models for elderly or frail individuals, such as the Canadian Programme of Research to Integrate Services for Maintenance of Autonomy (PRISMA) [4], which is intended to preserve functional independence among frail elderly people and reduce burnout syndrome among caregivers. The PRISMA model focuses on case management and coordinated care provision, connecting different service providers and providing a single access point for the healthcare system. 

Kaiser Permanente [5] has utilised a population stratification process to identify four population groups: (1) the general population, where the focus is on health promotion and disease prevention; (2) chronic patients (70–80% of the population), where the focus is on encouraging self-management and self-care support; (3) high-risk patients (15%), where the focus is on illness management; and (4) highly complex patients (5%), where case management services are prioritised. 

In Spain, the *Estrategia para el Abordaje de la Cronicidad* (Strategy for Tackling Chronicity) was published in 2012 [6]. In the years that followed, the country’s autonomous communities adapted the national strategy to regional population characteristics and healthcare systems. The Spanish chronic disease care strategy includes organisational changes required to manage chronic conditions, which are characterised by multimorbidity, comorbidity, and polypharmacy. The need for population stratification is highlighted by the COVID-19 public health emergency, with difficulties coordinating between care providers or healthcare services, economic impacts, and premature morbidity and mortality [7]. Therefore, implementing chronic care models and reorganising services to suit people and their needs remains a priority. It is vital that these models feature an approach that is focused on care rather than on disease. Many risk stratification models identify older adults with functional limitations in their daily lives who require significant health and social resources. 

The *Estrategia* highlights the need to perform population risk stratification to determine care needs targeted to personalised interventions. Population stratification to determine the level of complexity using the Kaiser pyramid must be combined with a comprehensive assessment of patients’ medical, care, functional, and social needs and planning of personalised interventions based on these needs, identifying the most appropriate resource for each clinical situation and stage of disease and drawing on available social and family support. 

Nurses play a central role in implementing the *Estrategia* in Spain. To record their PC activity, they use a specific assessment module in the electronic health record (EHR) based on Gordon’s functional health patterns [8]. This enables a comprehensive assessment of chronic, vulnerable patients with more extensive care needs by assessing the patient’s functional status in 11 assessment areas from a biopsychosocial perspective: (1) health perception/health management; (2) nutrition/metabolism; (3) elimination; (4) physical activity/exercise; (5) sleep/rest; (6) cognition/perception; (7) self-perception/self-concept; (8) role/relationships; (9) sexuality/reproduction; (10) coping/stress tolerance; and (11) values/beliefs. Altered health patterns are used to diagnose care needs or inform nursing diagnoses, which are classified using the standardised nursing language NANDA-I [9]. Meanwhile, health outcomes and care interventions are classified using the Nursing Outcomes Classification (NOC) [10] and Nursing Interventions Classification (NIC) [11], respectively. Nursing records in EHRs meet basic quality requirements as they are standardised under a single assessment framework (Gordon’s health patterns) and a system of standardised nursing languages: a nursing diagnoses (NANDA-I) classification, a health outcomes (NOC) classification, and a care interventions (NIC) classification. 

Spanish legislation on the data to be included in clinical records in the National Health System (Royal Decree 1093/2010 of 3 September) stipulates that standardised nursing languages must be used by nurses to record care activity in EHRs [12]. The use of standardised nursing languages in PC settings with chronic patients has been studied on several occasions, revealing benefits in terms of quality of information (clarity and understandability), attainment of health outcomes [13,14,15,16], and consumption of healthcare resources [17].

The hypothesis explored in this study is that the information recorded by nurses in EHRs, using an assessment framework based on health patterns and standardised nursing languages (NANDA-I, NIC, and NOC), is useful in describing the profile of highly complex chronic patients (HCCPs) as regular users of PC services.

The objectives of this study were as follows: (1) to describe the sociodemographic and clinical profile of HCCPs in a specific healthcare area; (2) to determine the level of completion and the functional status outcomes of the nursing assessment recorded on the EHR in PC; (3) to identify the most prevalent nursing diagnoses, planned outcomes, and associated interventions; and (4) to confirm the association between dysfunction in each health patterns, nursing diagnoses, and interventions, and the sociodemographic and clinical characteristics of HCCPs.

The study is intended for use by PC professionals, identifying assessment areas displaying frequent dysfunction among HCCPs, related care needs, health outcomes, and the interventions most commonly planned by nurses in EHRs. This information may also be used by managers in order to target care models to address the needs of high-risk, high-need patients.

## 2. Materials and Methods

### 2.1. Design and Sampling Method 

An observational, descriptive, cross-sectional, epidemiological study design was used. To conduct the study, the information recorded on the EHRs for all HCCP users at PC facilities in the Tenerife Healthcare Area (Canary Islands, Spain) on 30 September 2020 was used. The sample comprised a total of 51,374 patients. 

### 2.2. Study Setting

The Tenerife Healthcare Area is one of seven healthcare areas in the Canary Islands Health Service (SCS), which is part of Spain’s National Health System and delivers public healthcare services to the population of the Canary Islands. In 2020, the total population of this autonomous community was approximately 2,173,000, with around 929,000 people living on the island of Tenerife. 

The Tenerife Healthcare Area is divided into 41 basic healthcare districts, with a total of 103 healthcare facilities staffed by 795 nurses delivering PC services. The EHR used at these facilities is Drago-AP [18]. To record their nursing activity in Drago-AP, nurses use a specific module featuring a semi-structured assessment based on Gordon’s health patterns, 2015–2017 NANDA-I nursing diagnoses [19], the outcome criteria from the 5th edition of the NOC classification [20], and the catalogue of interventions from the 6th edition of the NIC classification [21]. In this way, care plans for HCCPs contain an EHR with data enabling nurses to conduct a comprehensive assessment by organising patients’ responses and applying practice-based knowledge to facilitate the diagnosis of priority care needs, as several studies have recommended [22,23]. 

### 2.3. Variables

The variables in the database created for this study were grouped by type:Sociodemographic variables:
○Sex. ○Age, in years and by groups: under 65 s; 65–79; over 80 s. ○Basic healthcare district.Clinical variables:
○Classification of participants aged ≥65: independent; frail; dependent.○Percentile of complexity (Pc) of the HCCP: highly complex (Pc = 95–99.4); extremely complex (Pc ≥ 99.5).○Chronic health conditions as listed in the International Classification of Diseases (ICD-10). Use of healthcare resources:
○Number of visits to the PC physician in the past year.○Number of visits to the PC nurse in the past year.○Number of hospital admissions in the past year.Functional status:
○Total number of health patterns assessed, with information in the EHR from Drago-AP.○Assessment data for each health pattern: self-perceived health (very good, good, average, poor, very poor); therapeutic follow-up (adequate, inadequate); eating habits (adequate, partially adequate, inadequate); special diet (yes, no); skin alteration/lesion (yes, no); pressure injury (yes, no); urinary elimination problems (yes, no); bowel elimination problems (yes, no); physical exercise habit (active, partially active, inactive); mobility (normal, with problems); home help service (municipal, private, both, no help); dependency in basic activities of daily living (BADL) (yes, no); dependency in instrumental activities of daily living (IADL) (yes, no); falls in the past 6 months (yes, no); indoor or outdoor architectural barriers (yes, no); employment status (self-employed or employed worker, student, homemaker, unemployed, pensioner or retiree, disabled, other); sleep problems (yes, no); visual impairment (yes, no); hearing impairment (yes, no); level of education (no education, primary education, secondary education or vocational training, university education); anxiety (yes, no); expressed fear or concern (yes, no); self-dissatisfaction (yes, no); presence of social risk factors affecting health (yes, no); living alone (yes, no); has a caregiver (yes, no); is a caregiver for someone with dependency in BADL or IADL (yes, no); expressed sexual dysfunction (yes, no); reproductive dysfunction (yes, no); has experienced major life changes in the past two years (yes, no); ability to handle difficult situations (good, average, poor); financial concerns (yes, no).○Outcome of assessment for each health pattern: normal; risk of alteration; altered; non-assessable. Nursing diagnoses:
○Total number of NANDA-I nursing diagnoses recorded in the EHR at the time of the study.○Presence of each NANDA-I nursing diagnose: yes; no.Planned outcomes:
○Total number of NOC outcome criteria recorded in the EHR at the time of the study.○Presence of each NOC outcome criterion: yes; no.Nursing interventions:
○Total number of NIC interventions recorded in the EHR at the time of the study.○Presence of each NIC intervention: yes; no.

### 2.4. Data Collection Procedure

An anonymised database was created in an Excel file containing all the information needed to carry out the analysis in accordance with the study objectives. Each dataset was extracted from the EHRs employing an automated search program. Each EHR was coded alphanumerically, with each dataset assigned a code according to the order of inclusion in the database. Once it had been anonymised, the database was exported in Excel format to the IBM SPSS© v.25.0 statistical programme for cleaning and subsequent analysis.

### 2.5. Data Analysis

The nominal variables were expressed using the absolute frequency of their categories, while the scale variables were expressed using means and standard deviations (SDs) or median and percentiles 5 and 95 (Pc5-Pc95), depending on whether or not the normality criterion was fulfilled. The bivariate analysis to identify associations and differences was performed using the chi-squared test. The strength of the relationship was assessed using Cramér’s V to counter the effect size due to the large sample size. To assess the magnitude of this coefficient, Rea and Parker’s criteria were used [24]. All tests were two-tailed with an alpha significance threshold set at <0.050 and carried out using IBM SPSS© v.25.0 software. 

## 3. Results

### 3.1. Sociodemographic and Clinical Profile of HCCPs in the Tenerife Healthcare Area

The sample was 57.4% female, with a mean age of 73.3 (12.2). In terms of age group, 22.3% of the sample was under 65, 43.7% was between 65 and 79, and 34% was over 80. The distribution of HCCP patients by basic healthcare district in relation to the total ranged between 0.5% and 5.5%. Among the participants aged 65 and over (77.7%), 73.2% were classified in the EHR as independent, frail, or dependent elderly people. The remaining 26.8% were not classified as independent, frail, or dependent elderly people. Among the classified participants, 49% were independent, 34.5% were frail, and 16.5% were dependent. With regard to the degree of complexity, 89.2% were HCCPs (Pc = 95–99.4) and the rest (10.8%) were extremely complex chronic patients (Pc ≥ 99.5). The prevalence of the most common chronic health conditions (greater than 20%) is shown in Table 1. 

The mean number of visits to the PC physician in the year preceding the study was nine (seven), while the mean number of visits to the PC nurse was five (eight). Moreover, 1.1% of the sample had not visited their physician or nurse in the past year, 87.9% had not been admitted to hospital in the year preceding the study, 8.7% had been admitted once or twice, and 3.4% had been admitted three times or more. 

### 3.2. Level of Completion and Functional Status Outcomes of Nursing Assessment

Some 14.8% of the sample had no health patterns assessed, while 49.2% had between one and four. Only 6.9% of the cases studied had records of assessment for all 11 health patterns. As for recording functional status outcomes, the frequency of health patterns was distributed as follows, from highest to lowest level of completion: 1 (70.9%), 2 (60%), 4 (55%), 3 (45.1%), 5 (39.9%), 6 (36.4%), 8 (21.5%), 7 (17.5%), 11 (8.7%), 10 (12.8%), and 9 (13.5%). 

Table 2 shows the functional status outcomes for each health pattern where information had been recorded. The highest percentage of dysfunction was observed in health patterns 6, 7, and 4 respectively, while the lowest percentage was observed in 9 and 11. 

The characteristics of the HCCP population, according to the assessment data for each health pattern, are shown in Table 3. 

### 3.3. Most Prevalent Nursing Diagnoses, Planned Outcomes, and Interventions

Up to 125 nursing diagnoses had been recorded on the Drago-AP EHR for the HCCPs studied. The mean number of active nursing diagnoses per patient was 7.3 (5.2), 1.4% had no nursing diagnoses recorded, 42.5% had between 1 and 5 nursing diagnoses on their EHR, and 35.7% had between 6 and 10.

With regard to planned health outcomes, up to 117 NOC outcomes had been recorded. The mean number per patient was 5.1 (4.1), 5.3% had no planned outcomes, 59.1% had between 1 and 5 NOC outcomes, and 25.7% had between 6 and 10.

With regard to the nursing interventions recorded, a total of 147 NIC interventions was observed with a mean of 8.1 (6.9), 4.4% of participants had no interventions on their EHR, 38.8% had between 1 and 5, and 29.4% had between 6 and 10 interventions.

The 15 most prevalent nursing diagnoses, NOC outcomes, and NIC interventions are shown in Table 4.

### 3.4. Associations between Dysfunction, Nursing Diagnoses and Interventions, and Patient Characteristics

Moderately and highly significant differences were observed between dysfunction in health pattern 4 and age group, and between dysfunction in health patterns 1, 3, 4, and 6 and classification as an independent, frail, or dependent elderly person (Table 5). Weakly significant differences were observed between dysfunction in health pattern 4 and physical exercise habit, and between dysfunction in health pattern 6 and chronic dementia.

With regard to the presence of certain nursing diagnoses, significant differences were observed by age group, classification of elderly person’s status, and presence of diseases such as dementia, depression, diabetes mellitus, asthma, chronic obstructive pulmonary disease and urinary tract infection (Table 6). 

Finally, a total of 20 NIC interventions showed moderately or relatively strong associations for older age groups, higher levels of dependency, and chronic health conditions such as asthma and dementia (Table 7).

## 4. Discussion

The study sample was mostly women (almost 60%), with a high mean age: one-third were aged 80 or over and approximately 40% were between 65 and 79 years old. The *Estrategia para el Abordaje de la Cronicidad* in Spain [6] describes population ageing and sets out projections for levels of dependency in the country over the coming decades, with chronic health conditions on the rise and more than 50% of people aged over 65 predicted to be dependent by 2050. In the Canary Islands, almost a quarter of the adult population has developed a chronic condition, and these conditions are most prevalent among women. This results in more extensive use of healthcare resources and points to the need to shift from a culture of healing to a culture of caring [25], as indicated in the *Estrategia de abordaje de la cronicidad en la Comunidad Autónoma de Canarias*. Although the Canary Islands has a younger population than the national average for Spain, the population is ageing more quickly than in the rest of the country. In our study, almost a quarter of the study population was aged under 65, which confirms that there are large numbers of people with highly complex chronic conditions at early ages who would benefit from a case management care model, as shown in other studies conducted in the same geographic and healthcare setting [26]. 

Just over half of the study population was classified as frail or dependent, highlighting the multiple care needs among HCCP users of these characteristics. However, the fact that the other half of the participants were independent suggests that they are able to participate in and take responsibility for their own care, aiming to remain independent in their daily lives, quality of life, and decision making. 

High blood pressure was the most prevalent chronic condition, echoing studies with similar characteristics, such as [26], followed by hyperlipidaemia, osteoarthritis, and diabetes mellitus, which were present in more than half of the sample. The potential care needs deriving from these conditions include an increased risk of cardiovascular disease and functional limitations. Almost 40% of the participants had been diagnosed with cardiac issues such as dysrhythmia and obesity. Four of the six most prevalent chronic conditions among the HCCP population (Table 1) are diagnostic criteria for metabolic syndrome. The prevalence of this syndrome in Spain is 31% [27], and research has shown that the degree of control of cardiovascular risk factors is very low among people at high risk. Therefore, recommended interventions are aimed at improving adherence to the therapeutic plan and encouraging a healthier lifestyle [28]. 

Other common chronic health conditions in our sample were depression and anxiety, which were present in almost one-third of participants. The study data were taken from EHRs written up in September 2020, six months after the SARS-CoV-2 coronavirus outbreak was declared a pandemic by the WHO [29]. The COVID-19 pandemic is known to have had an impact on mental health, leading to a significant increase in depression and anxiety [30,31,32]. Given the characteristics of this study, it was not possible to establish a cause–effect relationship in this regard. However, it is relevant to consider the recommendations made by experts on tackling these issues [33], as chronic depression and anxiety are highly prevalent in the Canary Islands, especially among women [25]. 

As for the average number of visits to PC professionals, including both physicians and nurses, the study population was found to use these services regularly, facilitating case management by providing more extensive information about the patients. Only a small proportion (1.1%) of the sample had not been seen by either of the aforementioned professionals in the past year. To improve uptake, several different chronic care strategies and models [1,2,3,4,5,6] [25] recommend that healthcare personnel take an active role and proactively identify people who would benefit from this approach to care. Meanwhile, approximately 10% of the sample had been admitted to hospital at least once in the past year. These cases are likely to require greater coordination between professionals, services, and care providers in order to ensure an adequate perceived quality of life and manage family, social, and healthcare resources.

With regard to the level of completion of EHRs in the assessment module based on health patterns, there was a lack of records in numerous areas. In the case of complex patients who are frequent users of PC services and require an adequate case management approach, this lack of information is an important area for improvement among the professionals who provide care for these users, i.e., nurses. Clinical information systems are a central component of chronic care models, as are PC teams. Therefore, discontinuity of information is an organisational issue that leads to discontinuity in overall care [6,25]. The HCCP population is a vulnerable group with numerous care needs, who need sufficient information to be recorded in the EHR to allow appropriate, early intervention to be planned. Inadequate recording of health patterns containing psychosocial information is particularly problematic, as identified in previous studies in the same healthcare area [34,35]. Nurses tend to record information in one, two, three, or even four health patterns, with this practice representing almost half of all cases. Functional status is most frequently recorded in the following health patterns: health perception/health management, nutrition/metabolism, physical activity/exercise, elimination, and sleep/rest (health patterns 1–5). This may be due to the fact that nurses find it easier to identify issues in biological and functional areas than in other, more psychosocial areas (health patterns 6–11). Based on these difficulties, which have been identified in previous studies, questionnaires have been compiled and validated to enhance the content of nursing assessments in general terms and in the psychosocial area, with a view of improving the diagnosis of care needs [36,37,38,39]. Professional mentoring programmes to train PC nurses in diagnostic assessment and recording and in care planning using standardised nursing languages in EHRs [40,41,42] have also been implemented with satisfactory results. However, in view of the results of this study, it seems advisable to reinforce a training strategy that focuses on the proper completion of EHRs. 

Broadly speaking, the assessment data by health patterns found in the EHRs (Table 3) indicates that HCCPs have an average, poor, or very poor self-perceived health; have adequate therapeutic follow-up and eating habits; have no special diet, skin problems, or elimination problems; are physically active or partially active; have no mobility issues or home help; have indoor architectural barriers; are retired; have visual and sleep problems; have a primary education; have a primary caregiver; have experienced major life changes in the past two years; and have an average or poor ability to handle difficult situations. 

When it comes to identifying possible care needs, the following aspects recorded during the assessment are particularly relevant: almost 30% of patients had inadequate or partially adequate eating habits; almost 40% followed a special diet; one-fifth of the sample had skin alterations/lesions and bowel elimination problems; more than one-third had urinary elimination problems; 70% of participants were inactive or partially active in terms of physical activity; almost one-third had mobility issues; 80% had no home help; one-quarter were dependent in BADL and IADL; one-fifth had suffered a fall in the past six months; almost 60% had architectural barriers in their homes, while just over 40% also had outdoor barriers; almost 90% were pensioners, retirees, homemakers, or unemployed; half had sleep problems; almost 80% had visual impairment, while almost one-third had hearing impairment; almost 90% had no education or only primary education; 40% suffered from anxiety and 50% from fear; one-fifth were not satisfied with themselves; just over 40% had social risk factors; more than one-quarter lived alone; half of the sample had a primary caregiver; 60% had undergone major life changes in the past two years; more than half had an average or poor ability to handle difficult situations; and 40% were worried about their financial circumstances. 

According to the results recorded for each health pattern (Table 2), the highest level of dysfunction was observed in the cognition/perception health pattern, with almost half of the study population affected, followed by the self-perception/self-concept, physical activity/exercise, elimination, and sleep/rest health patterns, which were altered in more than one-third of these patients. This profile allows the care needs and interventions required by the HCCP population to be identified. Our study aims to inform community care planning based on the health needs of patients with highly complex chronic conditions, as the Spanish strategy for tackling chronicity and several other studies have recommended [6,43]. In this approach, the curative model places little value on the care perspective that is essential to attain positive health outcomes. 

With regard to the number of nursing diagnoses recorded, the mean number observed was seven. Almost 40% of patients had between one and five. As a clinical statement, nursing diagnoses represent a specific care need. Given the complex nature and multiple needs of the study population, these figures seem reasonable. As for the types of problems diagnosed, the results echo those of other studies conducted in similar populations and healthcare settings [13]. If we group the nursing diagnoses by domain on the NANDA-I taxonomy [44], we find that 80% of the most prevalent in the HCCP study population (Table 4) belong to four of the thirteen domains: *Health promotion* (28%), *Activity/rest* (24%), *Safety/protection* (16%), and *Comfort* (12%). The health patterns most commonly identified as dysfunctional, which were listed above, are related to the same NANDA-I domains as the most prevalent issues. 

Spain’s *Estrategia para el Abordaje de la Cronicidad* recommends conducting comprehensive assessments in order to stratify the population and identify those at the greatest risk of falling ill [6]. This will allow care needs for chronic health problems to be predicted and interventions to maintain or improve quality of life to be planned. In our view, this study contributes to implementing these recommendations. Moreover, the skills listed in the regulations on Family and Community Nursing in Spain [45] include identifying population health needs and responding effectively to people with prevalent chronic conditions, disability, risk of disease, and frailty, as well as operating sentinel surveillance systems. standardised nursing languages, especially the NANDA-I classification, are useful tools for identifying patients’ care needs and recording the relevant information on the EHR. Nursing diagnoses are useful for the purposes described, and their validity as explanatory indicators in nursing service delivery in PC [46,47] and as predictors of healthcare resource consumption by patients [17] has been confirmed. 

With regard to planned health outcomes and nursing interventions associated with HCCPs in PC, the study shows a mean number of outcomes assigned in the EHR and planned interventions that does not appear to be excessive. This aspect could play a key role in implementing a functional, viable care plan. Taking the specific NOC outcome criteria into consideration, 84% of the most frequent outcome criteria belong to only two of the seven domains in the NOC classification [48]. The two domains in question are *Health knowledge and behaviour* (48%) and *Physiological health* (36%). This clearly points to a style of care planning for HCCPs that focuses primarily on health objectives describing attitudes, understanding, and specific actions relating to health and disease on the one hand, and objectives describing organ function on the other. It is relevant to note that none of the most prevalent planned outcomes belongs to the domain of *Psychosocial health*, *Family health,* or *Community health*. Like NANDA-I and NOC, the NIC also features different domains [49]. There are seven domains in the standardised nursing interventions classification. Among the most frequent interventions identified in this study, 84% belonged to three domains of the NIC: *Physiological: basic* (28%); *Physiological: complex* (28%), and *Safety* (28%). This type of care provides support for physical function, homeostatic regulation, and protection against possible harm. Care in the domains of *Family* and *Community* is conspicuously absent from the most prevalent interventions. This may be due to the fact that the approach to specialist family and community nursing, with specific competencies adapted to these levels rather than exclusively to the individual, remains undeveloped. 

In summary, with regard to care planning using standardised nursing languages for HCCPs in PC, there is significant dysfunction in the psychosocial domain, especially in health patterns 7, 10, and 8 (Table 2). Even so, problems (nursing diagnoses), outcomes, and interventions focus largely on physiological, functional, and behavioural aspects to minimise risk and ensure patient safety, with family and community aspects given lower priority. The implementation of people-centred chronic care models requires a change of approach to incorporate a genuine focus on family and community care in PC and achieve the threefold model recommended by various authors: personal, family, and community [50]. However, we are aware that recording individual clinical histories on the EHR can hinder or complicate the recording of family and community assessments, diagnoses, and interventions. 

Regular use of standardised nursing languages, which form part of the EHR in Spain’s National Health System, will provide more extensive knowledge of care and facilitate emerging lines of research on the care needs of vulnerable groups such as HCCPs [51]. This study was carried out with this recommendation in mind. 

The results of the bivariate analyses revealed significant differences between the level of dysfunction in certain health patterns (1, 3, 4, and 6) and the age group, classification of the elderly person’s status, physical activity, and cognitive issues variables (Table 5). These differences were to be expected, as our hypothesis suggested, although no evidence of them had been found in our study setting previously. This finding provides new knowledge on the use and benefits of standardised nursing languages in the EHR for HCCPs. 

The bivariate analyses also showed significant differences with regard to the assignation of certain nursing diagnoses, which were more frequent according to the age group, classification of the elderly person’s status, and presence of specific diseases (Table 6). This facilitates the diagnostic validation of the NANDA-I classification, as the organisation itself suggests [52], using large samples of specific populations such as HCCPs. In this way, knowledge can be generated in two specific, recommended types of epidemiological study on the language of nursing diagnoses: (1) identification of problems by frequency of occurrence in nursing settings, and (2) cross-sectional studies to identify their prevalence [53]. Significant, important differences were found with regard to the presence of a group of five nursing diagnoses relating to age and level of dependency: Impaired walking (00088), Impaired physical mobility (00085), Bathing self-care deficit (00108), Dressing self-care deficit (00109), and Risk for falls (00155). These diagnostic criteria are indicators in another nursing diagnosis that is part of NANDA-I: Frail elderly syndrome (00257). This may explain the profile and characteristics of the HCCPs studied. Another four (Anxiety—00146, Risk for unstable blood glucose level—00176, Ineffective breathing pattern—00032, and Impaired urinary elimination—00016) were significantly higher in frequency and strength of association when present alongside depression, diabetes mellitus, asthma, chronic obstructive pulmonary disease, and urinary tract infection, respectively. 

Finally, the statistical tests and bivariate analyses conducted with the NIC interventions and patients’ characteristics revealed significant associations between 20 of the interventions and age group, level of dependency, and presence of chronic conditions such as asthma and dementia (Table 7). These interventions relate to basic care for maintaining daily functional activity and personal safety. This care is fundamental for HCCPs and is included in the scope of competence of PC nurses [45], which recommends activities aimed at health promotion and disease prevention in vulnerable populations such as elderly people, or groups with a high degree of complexity. 

This study has several limitations. Firstly, the observational, descriptive, cross-sectional methodology used does not allow cause–effect relationships to be confirmed. This would require a cohort study design to monitor cases over time. However, due to the sample size, which covers the entire HCCP population in the healthcare area under study, this study is also epidemiological in nature and thus holds great descriptive value, allowing the characteristics of a specific group of highly complex patients to be explored from a clinical and care perspective. The large sample size may be viewed as a second limitation, as significant effects found in studies with a large sample size may be irrelevant. Therefore, significance tests are insufficient in practical situations where the magnitude of the observed effect is crucial. To mitigate this limitation, effect size procedures are used to quantify the relevance of the effect. This is why the strength of the relationship between variables was evaluated in this study using Cramér’s V and Rea and Parker’s criteria [24] to calculate the magnitude of the coefficient. The third potential limitation was that the study was carried out using information extracted from EHRs rather than information taken directly from nursing assessments to obtain all the necessary data about the patients. Research has shown that nurses regularly record fewer activities than they carry out in practice [54]. This is apparent from our results, which point to a lack of recording in many of the assessment areas by health patterns. It would be helpful for future training on the use of nursing languages in PC to focus on this aspect in order to obtain sufficient information about HCCPs in EHRs. On the other hand, we do not know how skilled the nurses who completed the electronic records were in applying the nursing process to real nursing practice or their knowledge of standardised nursing languages. This may lead to a limited use of the nursing terminologies, restricted to the most well-known, widely used elements of the NANDA-I, NIC, and NOC classifications, which contain a wide range of nursing diagnoses, health outcomes, and interventions. However, the study also has strengths. No previous epidemiological study describing the characteristics of HCCPs in the community setting from a care perspective using standardised nursing languages has been identified. This allows us to supplement and enrich the information available on the EHR and develop nursing interventions in PC that help improve care and quality of life for these people and their families. 

## 5. Conclusions

This study aimed to describe the sociodemographic and clinical profile of HCCPs in a specific healthcare area, care needs, expected outcomes, and planned nursing interventions in PC using standardised nursing languages. The degree of completion of the EHR with information relating to this group of users was explored, pointing to a lack of recording in numerous areas of interest. The most prevalent nursing diagnoses, health outcomes, and interventions in the PC setting were identified, and the associations between variables revealed correlations and differences that were anticipated but had not previously been evidenced in this study setting. Associations were found between the dysfunction of various health patterns and the age group or the frailty/dependency status of the elderly person. Relationships were also observed between the presence of certain nursing diagnoses and older age, frailty and dependency, and chronic conditions. In addition, a group of care interventions were associated with these characteristics of HCCPs. 

In light of the results obtained in this research, we suggest recommendations for practice that focus mainly on two aspects: firstly, the improvement of records and their completion with regard to nursing care, which may require further training and awareness raising in this regard; secondly, the opportunities arising from our results in identifying care needs at the community level, which may help to plan specific multidisciplinary interventions by PC teams, including requests for coordination in health and social care.

## Figures and Tables

**Table 1 healthcare-10-02512-t001:** Prevalence of health conditions in highly complex chronic patients.

Chronic Health Condition	Frequency (%)	*n*
High blood pressure	87.2	44,810
Hyperlipidaemia	80.0	41,105
Osteoarthritis	67.8	34,821
Diabetes mellitus	56.1	28,796
Dysrhythmia	39.2	20,133
Obesity	38.7	19,900
Depression	35.7	18,336
Anxiety	34.0	17,462
Asthma	29.4	15,082
Thyroid disorder	28.7	14,727
Cardiac conduction disorder	27.2	13,966
Chronic renal failure	27.1	14,060
Ischaemic heart disease	25.1	12,876
COPD *	23.8	12,232
Urinary tract infection	23.6	12,108
Heart failure	22.8	11,733
Prior neoplasm	20.5	10,531

* Chronic Obstructive Pulmonary Disease.

**Table 2 healthcare-10-02512-t002:** Functional status of the HCCP population according to outcomes of assessment by health pattern.

Health Pattern	Normal *		Risk of Alteration **		Altered ***		Non-Assessable ****	
	%	*n*	%	*n*	%	*n*	%	*n*
1. Health perception/health management	46.6	16,970	25.9	9425	27.1	9862	0.5	183
2. Nutrition/metabolism	46.0	14,169	22.6	6969	31.0	9566	0.4	109
3. Elimination	52.6	12,187	12.3	2851	34.8	8052	0.3	76
4. Physical activity/exercise	44.8	12,678	19.6	5533	35.2	9960	0.4	100
5. Sleep/rest	50.6	10,371	14.8	3026	34.2	7012	0.4	83
6. Cognition/perception	34.8	6510	16.7	3123	47.8	8948	0.8	141
7. Self-perception/self-concept	42.4	3813	17.2	1541	35.3	3174	5.1	455
8. Role/relationships	56.6	6253	19.4	2143	23.5	2601	0.5	52
9. Sexuality/reproduction	78.5	5467	4.3	297	6.2	429	11.0	767
10. Coping/stress tolerance	51.4	3394	17.4	1151	27.3	1800	3.8	253
11. Values/beliefs	74.0	3311	4.9	219	7.4	332	13.7	613

* Normal: Health pattern whose outcome is functional based on the nursing assessment. ** Risk of alteration: Vulnerability identified in the assessed health pattern. *** Altered: Health pattern whose outcome is dysfunctional based on the nursing assessment. **** Non-assessable: Health pattern whose content cannot be addressed in the nursing assessment due to the patient’s condition.

**Table 3 healthcare-10-02512-t003:** Characteristics of highly complex chronic patients according to assessment by health pattern.

Health Pattern	Information Collected during Assessment	Frequency	
		%	Total *n*
1. Health perception/health management	Self-perceived health: Poor, or very poor Therapeutic follow-up: Inadequate	50.4 7.2	30,918 26,420
2. Nutrition/metabolism	Eating habits: Inadequate Special diet: Yes Skin alteration/lesion: Yes Pressure injury: Yes	8.1 36.4 20.4 2.7	1589 32,848 22,753 24,834
3. Elimination	Urinary elimination problems: Yes Bowel elimination problems: Yes	35.7 22.2	21,405 3504
4. Physical activity/exercise	Physical exercise habit: Inactive Mobility: With problems Home help service: No help Dependency in BADL *: Yes Dependency in IADL **: Yes Falls in the past 6 months: Yes Indoor architectural barriers: Yes Outdoor architectural barriers: Yes Employment status: Pensioner or retiree	33.3 31.0 81.0 24.2 28.0 19.5 57.3 42.7 71.8	5639 27,629 12,911 22,268 21,910 19,879 6388 6388 18,953
5. Sleep/rest	Sleep problems: Yes	50.7	20,525
6. Cognition/perception	Visual impairment: Yes Hearing impairment: Yes Level of education: No education Primary education Secondary education or vocational training University education	77. 28.3 27.6 59.7 9.5 3.2	16,340 17,467 11,243
7. Self-perception/self-concept	Anxiety: Yes Expressed fear or concern: Yes Self-dissatisfaction: Yes	39.1 49.4 20.4	9740 9082 641
8. Role/relationships	Presence of social risk factors: Yes Living alone: Yes Has caregiver: Yes Is a caregiver for someone with dependency: Yes	43.1 22.2 50.1 15.0	3319 11,872 8657 4058
9. Sexuality/reproduction	Expressed sexual dysfunction: Yes Reproductive dysfunction: Yes	10.7 7.2	3603 4365
10. Coping/stress tolerance	Major life changes in the past two years: Yes Ability to handle difficult situations: Poor	62.5 12.7	6938 6236
11. Values/beliefs	Financial concerns: Yes	39.8	1541

* Basic activities of daily living (BADL). ** Instrumental activities of daily living (IADL).

**Table 4 healthcare-10-02512-t004:** Prevalence of NANDA-I nursing diagnoses, NOC outcomes, and NIC interventions.

Nursing Diagnosis (Code)	Frequency		NOC Outcome (Code)	Frequency		NIC Intervention (Code)	Frequency	
	%	*n*		%	*n*		%	*n*
Readiness for enhanced health management (00162)	76.6	39,373	Immunisation behaviour (1900)	37.3	19,161	Vital signs monitoring (6680)	44.7	22,981
Readiness for improve immunisation status (00186)	43.9	22,542	Compliance behaviour (1601)	32.5	16,676	Health education (5510)	44.4	22,834
Acute pain (00132)	42.8	21,989	Risk control (1902)	30.1	15,474	Immunisation/vaccination management (6530)	39.5	20,308
Impaired skin integrity (00046)	42.6	21,860	Wound healing: secondary intention (1103)	19.3	9915	Medication administration: intramuscular (IM) (2313)	38.0	19,510
Impaired comfort (00214)	28.8	14,802	Pain level (2102)	19.2	9888	Risk identification (6610)	25.2	12,934
Ineffective breathing pattern (00032)	21.2	10,882	Adherence behaviour (1600)	15.9	8187	Wound care (3660)	21.4	10,997
Risk for falls (00155)	19.6	10,080	Wound healing: primary intention (1102)	15.2	7816	Patient contracting (4420)	20.4	10,462
Ineffective protection (00043)	19.2	9851	Pain control (1605)	14.6	7513	Exercise promotion (0200)	20.2	10,395
Impaired urinary elimination (00016)	18.3	9410	Urinary elimination (0503)	12.8	6563	Teaching: prescribed diet (5614)	20.1	10,327
Risk for unstable blood glucose level (00179)	17.9	9208	Personal well-being (2002)	12.4	6361	Nutritional counselling (5246)	19.2	9861
Nutritional imbalance: excess (00001)	16	8240	Self-management: diabetes (1619)	11.1	5715	Health screening (6520)	16.4	8402
Risk for injury (00035)	15.9	8167	Fall prevention behaviour (1909)	10.3	5291	Teaching: prescribed exercise (5612)	13.9	7137
Risk for infection (00004)	15.2	7809	Tissue integrity: skin and mucous membranes (1101)	9.5	4903	Infection control (6540)	12.9	6621
Health-generating behaviours (00084)	13.9	7161	Weight control (1612)	8.8	4532	Fall prevention (6490)	12.5	6409
Ineffective health management (00078)	13.5	6960	Self-care: ADL * (0300)	8.6	4438	Referral (8100)	12.3	6327

* ADL: Activities of daily living.

**Table 5 healthcare-10-02512-t005:** Tests of association between dysfunction by health pattern (HP) and patient characteristics.

HCCP * Population with Dysfunction (%) by Variable Category	Chi-Squared	Cramér’s V **	*p*-Value	Strength of Difference **
By age HP 4. Physical activity/exercise: <65 y.o.: 10.2%; 65–79 y.o.: 15.2%; ≥80 y.o.: 30.9%	2356.729	0.214	<0.001	Moderate
By classification of elderly person’s status HP 1. Health perception/health management: Independent: 13.1%; Frail: 27.9%; Dependent: 44.1%	2116.008	0.269	<0.001	Moderate
HP 3. Elimination: Independent: 12.3%; Frail: 23.4%; Dependent: 44.5%	2253.255	0.278	<0.001	Moderate
HP 4. Physical activity/exercise: Independent: 9.1%; Frail: 32.6%; Dependent: 66.3%	6323.614	0.465	<0.001	Strong
HP 6. Cognition/perception: Independent: 12.8%; Frail: 25.2%; Dependent: 45.1%	2223.307	0.276	<0.001	Moderate
By physical exercise habit HP 4. Physical activity/exercise: Active: 5.3%; Partially active: 10.5%; Inactive: 19.0%	157.367	0.167	<0.001	Weak
By presence of dementia diagnosis HP 6. Cognition/perception: Without dementia: 15.6%; With dementia: 43.4%	1693.035	0.182	<0.001	Weak

* HCCP: Highly complex chronic patient. ** Strength of association criteria by Cramér’s V value: negligible = between 0.00 and <0.10; weak = between 0.10 and <0.20; moderate = between 0.20 and <0.40; relatively strong = between 0.40 and <0.60; strong = between 0.60 and <0.80; very strong = between 0.80 and 1.00.

**Table 6 healthcare-10-02512-t006:** Tests of association between presence of nursing diagnoses and patient characteristics.

Prevalence of NANDA-I Diagnosis (%) by Variable Category			Chi-Squared	Cramér’s V *	*p*-Value	Strength of Difference *
**a. By age**						
**<65 y.o.**	**65–79 y.o.**	**≥** **80 y.o.**				
Impaired walking (00088)						
4.1%	7.2%	21.2%	2663.925	0.228	<0.001	Moderate
Bathing self-care deficit (00108)			3304.200	0.254	<0.001	Moderate
4.3%	7.3%	23.6%				
Dressing self-care deficit (00109)			2618.455	0.226	<0.001	Moderate
3.4%	5.8%	19.0%				
Risk for falls (00155)			5340.522	0.322	<0.001	Moderate
6.0%	13.0%	37.1%				
**b. By classification of elderly person’s status**						
**Independent**	**Frail**	**Dependent**				
Impaired physical mobility (00085)			2892.734	0.315	<0.001	Moderate
3.7%	13.9%	32.6%				
Impaired walking (00088)			3975.334	0.369	<0.001	Moderate
4.1%	18.9%	41.2%				
Bathing self-care deficit (00108)			7581.887	0.509	<0.001	Strong
2.5%	17.8%	56.6%				
Dressing self-care deficit (00109)			6734.277	0.480	<0.001	Strong
1.8%	13.2%	48.4%				
Risk for falls (00155)			5815.532	0.446	<0.001	Strong
9.8%	34.7%	64.4%				
**c. By other disease diagnosis**						
**Without dementia**	**With dementia**					
Bathing self-care deficit (00108)			2865.862	0.236	<0.001	Moderate
10.1%	41.3%					
Dressing self-care deficit (00109)			2771.850	0.232	<0.001	Moderate
8.0%	35.9%					
Risk for falls (00155)			2098.299	0.202	<0.001	Moderate
17.5%	50.0%					
**Without depression**	**With depression**					
Anxiety (00146)			1993.291	0.197	<0.001	Weak- Moderate
7.7%	21.3%					
**Without diabetes**	**With diabetes**					
Risk for unstable blood glucose level (00179)			7296.907	0.377	<0.001	Moderate
1.6%	30.7%					
**Without asthma**	**With asthma**					
Ineffective breathing pattern (00032)			3987.828	0.279	<0.001	Moderate
13.8%	38.8%					
**Without COPD ****	**With COPD**					
Ineffective breathing pattern (00032)			3144.798	0.247	<0.001	Moderate
15.5%	39.3%					
**Without UTI *** diagnosis**	**With UTI diagnosis**					
Impaired urinary elimination (00016)			2797.831	0.233	<0.001	Moderate
13.3%	34.6%					

* Cramér’s V: negligible = between 0.00 and <0.10; weak = between 0.10 and <0.20; moderate = between 0.20 and <0.40; relatively strong = between 0.40 and <0.60; strong = between 0.60 and <0.80; very strong = between 0.80 and 1.00. ** COPD: chronic obstructive pulmonary disease. *** UTI: urinary tract infection.

**Table 7 healthcare-10-02512-t007:** Tests of association between the presence of recorded NIC interventions and patient characteristics.

NIC Intervention	Associated Characteristic and Frequency (%) of Intervention			Chi-Squared	Cramér’s V *	*p*-Value	Strength of Difference *
Self-care assistance (1800)	Age group			2446.985	0.218	<0.001	Moderate
	<65 y.o.	65–79 y.o.	≥80 y.o.				
	2.3%	4.8%	16.4				
	Classification of elderly person’s status			4672.457	0.400	<0.001	Strong
	Independent	Frail	Dependent				
	2.0%	13.1%	38.8%				
Environmental management: safety (6486)	Age group			2301.525	0.212	<0.001	Moderate
	<65 y.o.	65–79 y.o.	≥80 y.o.				
	2.3%	4.8%	15.9%				
	Classification of elderly person’s status			2516.919	0.293	<0.001	Moderate
	Independent	Frail	Dependent				
	3.7%	14.5%	29.9%				
Fall prevention (6490)	Age group			3512.337	0.261	<0.001	Moderate
	<65 y.o.	<65 y.o.	<65 y.o.				
	3.7%	7.7%	24.3%				
	Classification of elderly person’s status			3818.504	0.361	<0.001	Moderate
	Independent	Frail	Dependent				
	5.8%	22.5%	44.1%				
**NIC intervention**	Classification of elderly person’s status						
	Independent	Frail	Dependent				
Self-care assistance: bathing/hygiene (1801)	1.3%	10.2%	31.3%	3847.843	0.363	<0.001	Moderate
Self-care assistance: dressing/grooming (1802)	0.8%	6.6%	24.6%	3256.106	0.334	<0.001	Moderate
Self-care assistance: feeding (1803)	0.8%	6.6%	24.6%	1579.300	0.232	<0.001	Moderate
Self-care assistance: toileting (1804)	0.6%	2.2%	10.3%	1300.702	0.211	<0.001	Moderate
Self-care assistance: transfer (1806)	0.1%	1.5%	7.9%	1233.180	0.205	<0.001	Moderate
Support system enhancement (5440)	2.0%	7.2%	15.8%	1237.629	0.206	<0.001	Moderate
Home maintenance assistance (7180)	1.6%	8.8%	18.2%	1667.168	0.239	<0.001	Moderate
Caregiver support (7040)	2.6%	6.9%	16.5%	1179.810	0.201	<0.001	Moderate
Positioning (0840)	0.8%	2.7%	10.8%	1206.877	0.203	<0.001	Moderate
Urinary incontinence care (0610)	2.9%	9.0%	22.3%	1810.499	0.249	<0.001	Moderate
Body mechanics promotion (0140)	1.1%	5.3%	12.9%	1195.738	0.202	<0.001	Moderate
Case management (7320)	1.3%	6.2%	19.6%	2158.689	0.272	<0.001	Moderate
Pressure ulcer prevention (3540)	0.3%	1.9%	12.1%	1906.057	0.255	<0.001	Moderate
Exercise therapy: ambulation (0221)	1.4%	7.3%	16.6%	1542.070	0.230	<0.001	Moderate
**NIC intervention**	*Asthma*						
	*No*		*Yes*				
Medication administration: inhalation (2311)	3.9%		16.9%	2499.389	0.221	<0.001	Moderate
**NIC intervention**	*Dementia*						
	*No*		*Yes*				
Cognitive stimulation (4720)	1.7%		14.8%	2165.956	0.205	<0.001	Moderate
Dementia management (6460)	0.5%		10.2%	2759.266	0.232	<0.001	Moderate

* Cramér’s V: negligible = between 0.00 and <0.10; weak = between 0.10 and <0.20; moderate = between 0.20 and <0.40; relatively strong = between 0.40 and <0.60; strong = between 0.60 and <0.80; very strong = between 0.80 and 1.00.

## Data Availability

The data presented in this study are available upon request from the corresponding author. The data are not publicly available due to privacy/ethical restrictions.

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
