# Peer review of "Nursing Diagnoses, Planned Outcomes and Associated Interventions with Highly Complex Chronic Patients in Primary Care Settings: An Observational Study"

_healthcare, 2022, doi:10.3390/healthcare10122512_

Round 1
Reviewer 1 Report
Dear authors, in this paper you describe results of an observational study regarding highly complex chronic patients (HCCP) by focusing on care in terms of patients’ health needs. Although the study is well conducted, my suggestion is to change title, using a title more immediately correlated to the type of study. Best regards
Author Response
Reviewer 1
Comments and Suggestions for Authors
- Dear authors, in this paper you describe results of an observational study regarding highly complex chronic patients (HCCP) by focusing on care in terms of patients’ health needs. Although the study is well conducted, my suggestion is to change title, using a title more immediately correlated to the type of study. Best regards
- The title has been changed to specify the type of study involved.
Reviewer 2 Report
Thank you for the opportunity to review this article. This is an impressive study looking at a significant number of individuals in one country.
Please remember to write the full names of all concepts before using abbreviations.
In your Intro, you describe Kaiser Permanente as a model. This is incorrect. It is a system that is well-known for the model they employ. Please rephrase this. It is likely a matter of semantics.
Results:
"Among the participants aged 65 and over (77.7%), 73.2% were classified as independent, frail, or dependent elderly people."
What were the other 22.3% classified as?
"The mean number of visits to the PC doctor in the year preceding the study was 9(7), 246 while the mean number of visits to the PC nurse was 5(8)."
What does the number in ( ) signify? It is unclear.
There were a large number of acronyms used in this paper which tends to lower the readability of the document unless your specific expertise is in the nursing domain, and maybe even in the European healthcare system. I would consider simplifying where possible to increase overall comprehension without losing the integrity of the terminology.
Author Response
Reviewer 2:
- Thank you for the opportunity to review this article. This is an impressive study looking at a significant number of individuals in one country.
- Please remember to write the full names of all concepts before using abbreviations.
- All abbreviations have been reviewed.
- In your Intro, you describe Kaiser Permanente as a model. This is incorrect. It is a system that is well-known for the model they employ. Please rephrase this. It is likely a matter of semantics.
- We have redrafted the sentence to describe Kaiser Permanente as an organisational system rather than a model (line 71).
Results: - Among the participants aged 65 and over (77.7%), 73.2% were classified asindependent, frail, or dependent elderly people.
- What were the other 22.3% classified as?
- We have clarified this point in the following paragraph, which replaces the previous one:
Among the participants aged 65 and over (77.7%), 73.2% were classified in the EHR as independent, frail, or dependent elderly people. The remaining 26.8% were not classified as independent, frail, or dependent elderly people. Among the classified participants, 49% were independent, 34.5% were frail, and 16.5% were dependent.
-The mean number of visits to the PC physician in the year preceding the study was 9(7), while the mean number of visits to the PC nurse was 5(8)."
- What does the number in ( ) signify? It is unclear.
- The number in brackets is the standard deviation, as explained in the methods section, in data analysis, to express the summary of each and every scale variable.
- There were a large number of acronyms used in this paper which tends to lower the readability of the document unless your specific expertise is in the nursing domain, and maybe even in the European healthcare system. I would consider simplifying where possible to increase overall comprehension without losing the integrity of the terminology.- - We have reduced the number of acronyms used in our article for readability. We have removed the following acronyms from the manuscript text: Standardised Nursing Languages (SNLs); Chronic Care Model (CCM); Kaiser Permanente (KP); Health Patterns (HPs); Nursing Diagnoses (NDs); Chronic Obstructive Pulmonary Disease(COPD); and High Blood Pressure (HBP).
Reviewer 3 Report
1. This manuscript describes a review of available medical records and the health service area to identify nursing diagnoses, nursing interventions pertinent to identifying high risk high need patients to aid in developing targeted patient-centric interventions.
2. The manuscript is too long, requires a more concise description, collapsing of table data, targeting of the discussion, and adding major study outcomes, policy and practice recommendations in the conclusion.
3. Abstract: Define NOC, NIC, provide major data findings (significant associations)
4. Make the introduction shorter and more concise: i.e. line52: identification of populations at risk for poor outcomes, line 70: :Kaiser Permanente has utilized a population stratification process to identify four population groups, line 83: The need for population stratification is highlighted by the COVID public health emergency..., line 88: Many risk stratification models identify older adults..., line 91: The Estrategia highlights the need to perform population risk stratification to determine care needs targeted to personalized interventions, line 122: ND and NIC , can be utilized for population risk assessment, line 128: to assess the association between dysfunction in each..., line 133: To target care models to address the needs of high-risk, high-need patients
5. Methods: Data Collection: were charts reviewed manually or was an automated EHR search program used?
6. Table 1: Collapse the categories and report the most common conditions i.e. greater than 20% prevalence
7. lines 245-249: Place this information in the table
8. Section 3.2: Do not begin sentences with numerals
9. Table 2: Define the categories: normal, risk of alteration, altered, not assessable in the legend
10. Tables 3, 4: much too long - collapse categories to ADL, IADL, caregiver concerns, educational level
11. Section 3.3: Place in a table
12. Tables 5, 6, 7: collapse the categories
13. Discussion: Consider shortening this making it more concise and focusing on A) Identification of ND, B) NIC creation following from A, C) Described the problem of incomplete records and describe ways to improve chart completion, D) how can you automate the identification of NOC and NIC and provide these findings to the collaborative health care team?
14. Conclusion: Detail significant outcomes (correlations), policy, and practice recommendations
Author Response
Reviewer 3
Comments and Suggestions for Authors
1. This manuscript describes a review of available medical records and the health service area to identify nursing diagnoses, nursing interventions pertinent to identifying high risk high need patients to aid in developing targeted patient-centric interventions.
2. The manuscript is too long, requires a more concise description, collapsing of table data, targeting of the discussion, and adding major study outcomes, policy and practice recommendations in the conclusion.
- We have shortened the manuscript and included more concise descriptions in accordance with the comments received from the reviewers.
3. Abstract: Define NOC, NIC, provide major data findings (significant associations)
- We have defined the NOC and NIC before using their acronyms and described themain results of the bivariate analysis (significant associations of note).
4. Make the introduction shorter and more concise: i.e. line52: identification of populations at risk for poor outcomes, line 70: Kaiser Permanente has utilized a population stratification process to identify four population groups, line 83: The need for population stratification is highlighted by the COVID public health emergency..., line 88: Many risk stratification models
identify older adults..., line 91: The Estrategia highlights the need to perform population risk stratification to determine care needs targeted to personalized interventions, line 122: ND and NIC , can be utilized for population risk assessment, line 128: to assess the association between dysfunction in each..., line 133: To target care models to address the needs of high-
risk, high-need patients
- We have included all the reviewer’s suggestions and modified the Introduction section accordingly, except for the recommendations associated with line 122, as the NIC classification of nursing interventions and the NOC classification of nursing outcomes cannot be used for population risk assessment, only the NANDA-I classification of careneeds can be used for that purpose.
5. Methods: Data Collection: were charts reviewed manually or was an automated EHR search program used?
- An automated EHR search program was used for this purpose (data collection). We have included a sentence in this section of the manuscript for clarification.
6. Table 1: Collapse the categories and report the most common conditions i.e. greater than 20% prevalence
- We have reduced the number of categories by removing chronic conditions with a prevalence below 20%.
7. lines 245-249: Place this information in the table
- Table 1 itself covers lines 245-249. However, if the reviewer means that the information in the previous paragraph should be placed in that table, we believe that this would not be appropriate, as they are different data sets. Therefore, we would rather leave the information in the previous paragraph as text, as it is presented.
8. Section 3.2: Do not begin sentences with numerals
- We have modified the beginning of the sentence so that it does not start with numerals.
9. Table 2: Define the categories: normal, risk of alteration, altered, not assessable in the legend
- We have defined the categories in the legend:
*Normal: Health pattern whose outcome is functional based on the nursing assessment;
**Risk of alteration: Vulnerability identified in the assessed health pattern; ***Altered: Health pattern whose outcome is dysfunctional based on the nursing assessment; ****Non-assessable: Health pattern whose content cannot be addressed in the nursing assessment due to the patient´s condition.
10. Tables 3, 4: much too long - collapse categories to ADL, IADL, caregiver concerns, educational level
- Table 3 shows the main functionality variables included in each of the eleven health patterns in M. Gordon’s assessment framework (1994), which are of great interest for our study and presentation of results. To limit ourselves to merely showing in this table information on four variables (ADL, IADL, caregiver concerns, and educational level) would be to oversimplify the data needed for a minimum nursing assessment. Nevertheless, in order to reduce the size of the table, we have removed a number of response categories toshow the most salient results for each variable.
- Table 4 now shows the prevalence of the 15 (not 25) most frequent NANDA-I, NOC, and NIC labels.
11. Section 3.3: Place in a table
- We believe that the information in this paragraph should not be in a table, given the large number of tables already in the manuscript and the fact that the message conveyed can easily be understood in text format.
12. Tables 5, 6, 7: collapse the categories
- Tables 5, 6, and 7 have been condensed in terms of format and reduced in size, making them easier to understand and without deleting any content categories.
13. Discussion: Consider shortening this making it more concise and focusing on A)Identification of ND, B) NIC creation following from A, C) Described the problem of incomplete records and describe ways to improve chart completion, D) how can you automate the identification of NOC and NIC and provide these findings to the collaborative health care team?
- It goes without saying that are very grateful for the recommendations made in this comment on the Discussion section. However, we consider it essential for our manuscript to maintain its present discursive form and structure, as we believe that this detailed analysis is one of the strengths of our work, despite its considerable length.
Reducing the size of the discussion would mean losing the consistency and commonthread that we have followed, which is structured in an analysis and contrast of resultsaccording to the following blocks or sub-sections: (a) sociodemographic and clinical data of patients, including data on resource use; (b) completion of electronic health records, specifically completion of the following sections: assessment (dysfunctionalityaccording to health patterns), diagnosis of care needs (NANDA-I diagnoses), and
planning of goals (NOC outcomes) and (NIC) interventions; (c) discussion/analysis of the results of bivariate tests; and (d) limitations. Therefore, we believe that it would not be appropriate to structure the discussion in the sub-sections proposed by the reviewer in this point. For instance, the suggested sub-section A) would start with the identification of nursing diagnoses, which would mean leaving out the entire sociodemographic and clinical framework/profile of highly complex chronic patients, as well as their functionality characteristics (collected in the assessments section by health patterns). This, in turn, would leave out the analysis and discussion of elementary questions about this patient profile. Regarding the suggested sub-section B), i.e. NIC creation following from A, we would like to clarify that NIC interventions are not created from nursing diagnoses, nor are NOC outcomes, and automation (as suggested by the reviewer for the sub-section D) of the NANDA-NOC-NIC process is strongly discouraged. Proposals have been made based on expert consensus in this respect, but automation of EHR terminologies is outside the scope of this study. Our aim is to describe the nursing care documented by nurses and to discuss our findings in a large
sample of highly complex chronic patients.
Regarding the proposed sub-section D), we think that the problem of incomplete records is sufficiently addressed between lines1304-1326 and also in the limitations section at the end of the discussion, lines 1503-1512. However, after line 1326, a sentence is added that explicitly provides a potential improvement in the case of incomplete EHRs.
14. Conclusion: Detail significant outcomes (correlations), policy, and practice
recommendations
- In this section, we have detailed the outcomes that obtained significant correlations and added a new paragraph on recommendations for practice and health policy based on the results of our study.
Round 2
Reviewer 3 Report
The authors have made credible responses to reviewer comments.